# Pursuing High-Resolution Structures of Nicotinic Acetylcholine Receptors: Lessons Learned from Five Decades

**DOI:** 10.3390/molecules26195753

**Published:** 2021-09-23

**Authors:** Manuel Delgado-Vélez, Orestes Quesada, Juan C. Villalobos-Santos, Rafael Maldonado-Hernández, Guillermo Asmar-Rovira, Raymond C. Stevens, José Antonio Lasalde-Dominicci

**Affiliations:** 1Department of Biology, Rio Piedras Campus, University of Puerto Rico, San Juan 00931, Puerto Rico; manuel.delgadovelez@upr.edu (M.D.-V.); juan.villalobos@upr.edu (J.C.V.-S.); 2Clinical Bioreagent Center, Molecular Sciences Research Center, University of Puerto Rico, San Juan 00926, Puerto Rico; quesada.orestes@gmail.com (O.Q.); rafael.maldonado1@upr.edu (R.M.-H.); 3Department of Biology, Arecibo Campus, University of Puerto Rico, Arecibo 00631, Puerto Rico; 4Department of Physical Sciences, Rio Piedras Campus, University of Puerto Rico, San Juan 00931, Puerto Rico; 5Department of Biology, Humacao Campus, University of Puerto Rico, Humacao 00792, Puerto Rico; 6Independent Researcher, Ambler, PA 19002, USA; guillermo_asmar@yahoo.com; 7Michelson Center for Convergent Bioscience, Department of Biological Sciences, Bridge Institute, University of Southern California, Los Angeles, CA 90007, USA; stevens@usc.edu; 8Department of Chemistry, Rio Piedras Campus, University of Puerto Rico, San Juan 00931, Puerto Rico; 9Institute of Neurobiology, Medical Science Campus, University of Puerto Rico, San Juan 00901, Puerto Rico

**Keywords:** nicotinic acetylcholine receptors, ligand-gated ion channel, *Torpedo californica*, *Torpedo marmorata*, cryogenic electron microscopy, crystal structure, detergents, lipids, crystallography, cholesterol

## Abstract

Since their discovery, nicotinic acetylcholine receptors (nAChRs) have been extensively studied to understand their function, as well as the consequence of alterations leading to disease states. Importantly, these receptors represent pharmacological targets to treat a number of neurological and neurodegenerative disorders. Nevertheless, their therapeutic value has been limited by the absence of high-resolution structures that allow for the design of more specific and effective drugs. This article offers a comprehensive review of five decades of research pursuing high-resolution structures of nAChRs. We provide a historical perspective, from initial structural studies to the most recent X-ray and cryogenic electron microscopy (Cryo-EM) nAChR structures. We also discuss the most relevant structural features that emerged from these studies, as well as perspectives in the field.

## 1. Introduction

The muscle-type nicotinic acetylcholine receptor (nAChR) is an integral membrane protein composed of five transmembrane subunits, with stoichiometry α2βγδ in a pentamer arrangement with an internal ion channel at its center. Each subunit is composed of four transmembrane segments (M1, M2, M3, and M4) located in the transmembrane region, and the N- and C-termini are located extracellularly. These overall structural features are shared with neuronal nAChRs, as well as the glycine (glyR), γ-aminobutyric (GABA_A_), serotonin (5-hydroxytryptamine (5-HT3)), and chloride-permeable glutamate receptors (GluRCl), all classified as ligand-gated ion channel membrane proteins [1,2,3]. In vertebrate neuromuscular junctions, binding of two acetylcholine (ACh) molecules to the two agonist-binding sites located at the nAChR’s α-γ and α-δ subunit interfaces [4,5,6] triggers conformational changes that allow activation and subsequent muscle contraction.

Since their discovery, nAChRs have been extensively studied to understand their function, as well as the consequences of alterations leading to several disease states. Thus, these receptors represent valuable pharmacological targets to treat various diseases and disorders that afflict humans. However, their therapeutic value has been limited by the absence of high-resolution crystallographic structures that allow for the design of more specific and effective drugs to treat diseases more efficiently. Structural enablement of ligand-gated ion channels has been extremely difficult. This research area has historically been constrained by continued challenges associated with the preparation of high-yield, monodisperse protein samples, an essential requirement for successful structural studies. The main source of nAChRs has been native sources (electric organ tissues from rays) or recombinant sources, either of which necessitates detergent solubilization and purification. During the last decade, but mainly within the last few years, important advances in cryogenic electron microscopy (Cryo-EM) technology has enabled structural determination of some nAChRs.

Here, we present a historical account of how nAChR crystallization has advanced to yield the first and only X-ray structure [7] of a neuronal nAChR, as well as the more recent Cryo-EM structures (Figure 1). We also comment on the historical challenges the field has faced trying to obtain high-resolution nAChR structures, outlining detergent solubilization and purification challenges, as well as the importance of cholesterol molecules for protein crystallization.

## 2. Historical Notes on nAChRs

The acetylcholine (ACh) molecule was first synthesized by Adolf von Baeyer in 1867 and later recognized as a parasympathetic nerve stimulator in 1914. It was eventually identified as a neurotransmitter at the skeletal neuromuscular junction in 1936. Later, in 1967, it was discovered that α-bungarotoxin (α-Bgtx), the active toxin present in the banded krait (Bungarus multicinctus) venom, specifically binds irreversibly to nAChRs [8], providing the opportunity to employ affinity chromatography to purify [9] and quantify nAChRs. Experimentally speaking, the main nAChR native sources were the Torpedo californica (Tc) and Torpedo marmorata (Tm) electroplaques from the electric organs of these animals, which are enriched in muscle-type nAChRs [10]. Subsequently, Steven G. Blanchard’s group performed the first partial protein sequencing on these muscle-type nAChRs [11], which were the first to be cloned [12,13].

For more than 50 years, nAChRs have been studied by spectroscopic, biochemical, computational, mutagenesis, X-ray, and Cryo-EM approaches, among others. This historical path began with the studies performed by Arthur Karlin in the 1960s, resulting in identification through affinity labeling, of the α1 subunit of muscle-type nAChRs isolated from electric rays [14]. These studies led to identification of the amino acids that contribute to the binding site for the natural agonist ACh using labeling reagents [15,16,17]. Additional studies performed by Patrick and others in 1973 led to the hypothesis that nAChRs were composed of several subunits, which were finally identified in the 1990s and named α, β, γ, and δ, and their increasing molecular weight [6,18]. Eventually, obtaining milligram quantities of purified nAChRs led to the understanding of the molecular basis of the autoimmune disease myasthenia gravis, and the implication of muscle-type nAChRs in this disease [19]. Advances in nAChR purification also allowed sequencing of the N-terminus of the four subunits [11], determination of the Tc nAChR subunit stoichiometry (α2βγδ) [20,21], awareness of subunit homology, and cloning of the cDNAs for the nAChR subunits. Cloning brought the nAChR into the molecular biology field, where the structural homology of all nAChR subunits was confirmed, providing access to a completely new series of novel technical approaches to investigate its structure and function. Furthermore, cloning of the cDNAs from other neurotransmitter-gated ion channels enabled sequence homologies to be identified, which prompted classification of nAChRs within the same gene superfamily [2,22,23].

During the last 40 years, great efforts have been made to determine nAChRs structure at the atomic level. The muscle-type 4.6 Å resolution Cryo-EM structure published by Nigel Unwin was the highest resolution structure until Brejc and co-workers were able to crystallize the soluble acetylcholine binding protein (AChBP) from Lymnaea stagnalis [24], a structural homolog of the extracellular domain of the muscle-type nAChRs. Brejc’s structure confirmed much data previously acquired by a plethora of methods over the years and has provided a starting point for a list of extracellular domain structures of different nAChRs. In actuality, for Homo sapiens, there are multiple extracellular domain structures of different nAChRs in the Protein Data Bank (PDB). Presently, the amount of structural data available on nAChRs is remarkable, and their rich history, physiological importance, structural, and clinical significance [25] make them one of the most interesting and rewarding membrane proteins to study.

## 3. nAChRs as Therapeutic Targets for Neurological Diseases

The crucial role that nAChRs play in the transmission of nerve impulses to and from the central nervous system (CNS) directly implicates them in various neurological diseases. These include, but are not limited to, congenital myasthenic syndromes (CMS), Alzheimer’s disease (AD), Parkinson’s disease (PD), schizophrenia, epilepsy, Tourette syndrome, and neuroinflammation, tobacco addiction among others, many of which afflict the aging segments of populations worldwide. Therefore, the importance of determining nAChR structural features at high resolution is clearly to design new and improved therapeutic treatments for the multiple disease in which these class of ion channels has been repeatedly implicated. Indeed, a high-resolution structure of the nAChR and its complexes with various nicotinic ligands is of crucial importance to treat the aforementioned diseases. In addition, elucidation of nAChR structures, at the peripheral level, will serve to design and test drugs to treat diseases of the neuromuscular junction such as myasthenia gravis, CMS, Lambert–Eaton myasthenic syndrome, botulism, slow-channel congenital myasthenic syndrome, and fast-channel congenital myasthenic syndrome, among others. Without question, the determination of such structures is critical for the field of neurology.

The heteromeric α4β2 nAChR and the homopentameric α7 nAChR are the most abundant nAChRs in the nervous system [26]. It has been shown that a significant loss of cholinergic neurons and acetylcholine receptors (~50%) occurs in the brain tissues of Alzheimer’s patients [27,28,29,30,31,32,33,34,35,36]. These two types of receptors are highly distributed in areas of the brain affected by Alzheimer’s disease, making these subtypes primary therapeutic targets for AD [37]. The rationale is that a drug that could enhance cognitive benefits and decrease progressive disease while protecting against amyloid β plaque formation could provide great benefits to AD patients. The use of selective agonist(s) or positive allosteric modulators could improve cholinergic function, through selective activation of the functional α7 and/or α4β2-nAChRs present in cholinergic pathways, which would enhance expression levels of other neurotransmitters such as dopamine, norepinephrine, and γ-amino butyric acid in the central nervous system (CNS).

The α7 nAChR has unique activation properties, including the highest calcium permeability among all nAChRs subtypes [37], as well as metabotropic action [38,39,40,41]. Moreover, α7 nAChR participation in a number of psychiatric and inflammatory disorders have made this nAChR subtype a highly relevant evolving drug target [37,42,43,44,45,46]. The recent cryo-EM structure of the α7 nAChR/epibatidine and PNU-120596 complex (PDB 7KOX, Table 1) represents a remarkable breakthrough, and a blueprint for the design of novel allosteric modulators that could enhance cognitive performance to treat many neurological and psychiatric conditions associated with deficits in cholinergic transmission. Moreover, the α4β2 nAChR is a primary target for nicotine’s addictive effects, and the binding site of the smoking cessation drug varenicline has been recently identified in a α4β2/varenicline complex (PDB 6UR8) (see Table 1).

Another remarkable example is AT-1001, a highly selective α3β4-nAChR ligand developed by Astraea Therapeutics as a potential treatment for drug addiction. AT-1001 has been shown to reduce nicotine withdrawal symptoms in rats [47]. Disruption of the α3β4 nAChR using highly selective drugs such as AT-1001 may interrupt this reward circuitry in the brain. The recent cryo-EM model for the α3β4/AT-1001 complex (PDB 6PV8, Table 1) has provided key structural information about the location of this ligand at its binding pocket, but also compares the binding site to the α4β2 receptor. Interestingly, AT-1001 adopted different orientations at the two binding sites [48,49].

Given that several nAChRs subtypes are distributed throughout many tissues, a ligand that binds nonselectively to several subtypes could produce collateral effects. Presently, the biggest challenge is to develop highly selective agonists, antagonists, and allosteric modulators for each nAChR subtype. The recent nAChRs structures will serve as a template to guide the development of highly specific nAChRs ligands.

## 4. Attempts to Crystallize the Muscle-Type nAChR

Early muscle-type nAChR structural and functional data were obtained from detergent-solubilized affinity-purified receptors obtained from nAChR-enriched membranes isolated from the electric organ tissues of Torpedo rays, such as *T. californica*, *T. nobiliana*, and *T. marmorata*. Over the past four decades, multiple attempts have been made to crystallize the nAChR [50,51]: Hucho’s group published small crystals but no X-ray diffraction data [50], while Stroud’s group obtained several X-ray diffraction patterns that were never reported. In parallel studies, our laboratory (www.nachrs.org (accessed on 21 September 2021)), in collaboration with Dr. Raymond C. Stevens laboratory at The Scripps Research Institute (TSRI, La Jolla, CA, USA), prepared Tc nAChR crystals using similar techniques and screened more than 50,000 crystallization conditions of several Tc nAChR preparations, using robotics combined with vapor diffusion methods (VDM), such as hanging drop and sitting drop. Unfortunately, these experiments yielded similar experimental outcomes to those of Hucho and Stroud: low-resolution X-ray diffraction that was not sufficient for meaningful structural studies. As a result, we opted to re-direct our research efforts towards probing the effects of different detergents on the stability, ion channel function and lipid composition of the Tc muscle-type nAChR. These studies revealed that the receptor is stable, functional and monodisperse in lipid-analog detergents, or in detergents that retained moderate amounts of residual native lipids, whereas the opposite was true about non-lipid analog detergents [52,53]. Over the past decade, we embraced the lipidic cubic phase (LCP) technique, as many other membrane protein laboratories have done, with the goal of enhancing our chances for successful nAChR structural studies. We performed crystallographic screening of about 5000 conditions for the Tc nAChR using the lipidic cubic phase (LCP) technique [54,55] and harvested a small number of crystals that in most cases yielded no diffraction at the GM/CA-CAT 23-ID-B/D beamline at Advanced Photon Source (https://www.aps.anl.gov/ (accessed on 21 September 2021)). Interestingly, there had been no published reports involving nAChR crystal structures since the late 1980s until the recent α4β2 X-ray structure [7].

Over the last two decades, several nAChR structural studies have been published where heterologous expression systems, like bacteria and yeast, have been utilized. Bacteria in particular lack the proper machinery for posttranslational modifications (i.e., glycosylation) and have a different membrane lipid composition by virtue of being prokaryotic organisms, which could give rise to nAChR structures that cannot replicate the receptor’s native state. In contrast, yeasts, as eukaryotic organisms, are capable of incorporating post-translational modifications, and have been used successfully for structural studies that have yielded significant advances in the field, including high-resolution X-ray structures of AChBP and mouse-α1 nAChR extracellular domain. Studies performed by Brejc and collaborators [24] produced the first high-resolution crystal structure of the homologous AChBP at 3.3 Å resolution; AChBP has ~24% sequence identity with muscle-type nAChR. In 2007, a high-resolution X-ray structure of the mouse-α1 nAChR extracellular domain, also expressed in yeast (Pichia pastoris), was solved at 1.94 Å resolution [56]. A large number of AChBPs (181,969 structures) from extracellular domains of different nAChRs from different species are now available in the Protein Data Bank (accessed 7 September 2021). Remarkably, the α2 extracellular domain is the only one that has crystallized as a pentamer, revealing a full ligand binding pocket conferred by two adjacent α2 subunits. Therefore, after several decades and huge research efforts from independent laboratories, the best approximation to a high-resolution structure of the nAChR was a collection of fragmented structures from extracellular domains from different species.

## 5. Early Cryo-EM Structural Studies of nAChRs

Over the course of several decades, Nigel Unwin and coworkers have published a series of studies describing the structural, physiological features, and characteristics of the Torpedo nAChRs. Typically, studies using electron microscopy (EM) used crude membranes from Tm electric tissues. Briefly, nAChR membranes were overlaid over carbon support grids and subsequently frozen in liquid ethane before the acquisition of EM images. nAChR crystals exhibiting round tube morphologies were then visualized via this freezing procedure. Cryo-EM images were further processed by selecting those showing the least degree of distortions, averaging multiple images using Fourier transforms. The use of Fourier transforms reduced the signal-to-noise ratios and increased the resolution significantly, enabling Unwin and coworkers to attain resolution values of up to 4.6 Å [57].

Three articles published during the 1990s described variations of the aforementioned experimental methods. The 1993's paper analyzed the EM images of the two-dimensional tubes to probe the structural features of muscle-type nAChR from Tm postsynaptic membranes at 9 Å resolution [58]. The 1993's paper used a similar sample preparation protocol, but imaged the nAChR in the open state by spraying the three-dimensional tubes with ACh before imaging [59]. A 1999 article employed multiple averaging of various tube types using Fourier transforms [57]. Importantly, the 1993's paper described in detail the structural features of the nAChR transmembrane regions using data obtained from the EM images. These experiments clearly identified two density areas in the EM images corresponding to the five α-helices of the M2 transmembrane segments of the nAChR subunits lining the ion-conducting pore. High-density areas of the images were identified as α-helices based on their dimensions, along with the packing and twisting pattern observed around one another, while the β-sheet regions were assigned based on their lack of significant, clearly defined features [58]. The low-density areas of the images corresponding to the M1, M3, and M4 transmembrane segments, were proposed to have a β-sheet secondary structure based on the lack of symmetry.

## 6. The Challenge of Crystal Formation for nAChRs

The success of protein structure determination using X-ray diffraction depends on protein crystal quality. Integral membrane proteins such as nAChRs, whose native environments are cell membranes, have proved to be extremely difficult to crystallize because detergent solubilization and purification, which irreversibly disrupts the native lipid environment, are necessary pre-conditions for membrane protein purification, crystal growth, and X-ray crystallography studies. Finding the right balance between partial preservation of the native lipid environment to retain the receptor’s biochemical integrity and functionality, versus sample purity and homogeneity, is an ever-present challenge for membrane protein structural studies. These challenges remain to this day in spite of significant advances like synthetic lipids and detergents, LCP, and Cryo-EM.

Further complicating matters is the fact that the natural biochemical environment of nAChR is known to be far from the high-salt concentration and aqueous chemical conditions that most crystallization reagents require to trigger the controlled precipitation processes that facilitate protein crystallization. Currently, the main challenges to nAChR crystallization are (1) sample purity, (2) heterogenic pentamers, (3) multiple stoichiometries, (4) pseudosimetry of heteropentamers, (5) extracellular domain glycosylation with heterogeneous glycan compositions, (6) large, disordered intracellular domains (M3-M4 loop), and (7) different receptor conformations (i.e., resting, conducting, desensitized, and blocked). In general, preparation of nAChRs and membrane protein crystals suitable for X-ray diffraction studies have become remarkably handcrafted experiments, which predictably leads to reproducibility issues, one of the foremost obstacles to attaining high-resolution structures.

In the previous pages, we have discussed in detail the main advantages and drawbacks of the numerous research approaches deployed for nAChR structural studies over the last four (4) decades. nAChR structural information has advanced considerably over that long period, but existing challenges have continued to slow down drug development for the treatment of numerous nAChR pathologies. In recent years, Cryo-EM has emerged as a viable alternative to X-ray crystallography for nAChR and more broadly integral membrane protein structural studies, because of its significantly lower purified protein yield requirements and the simple fact that protein crystal growth is no longer necessary. That being said, equipment cost hinders accessibility, as well as existing challenges related to data storage and resolution values that in some cases cannot match those obtained with X-ray crystallography. Table 1 is a summary of the methodologies used to obtain nAChR cryo-structures.

So far, the neuronal α4β2 nAChR is the only structure that has been solved using X-ray diffraction [7]. The structural data from this study was collected from “one crystal” out of thousands of crystals screened (personal communication with Dr. Claudio Morales-Pérez). Eventually, these difficulties of reproducing high-quality α4β2 nAChR crystals led the authors to use Cryo-EM as an alternate structural determination method in their studies. In 2018, the same group determined two different stoichiometries of the same α4β2 nAChR using Cryo-EM [48]. Although these two α4β2 nAChR structures have provided key information about nicotine and cholesterol-binding, subunit stoichiometry, and oligomerization, both are low-resolution structures (~3.9 Å). Therefore, the field still lacks high-resolution nAChR structures.

This year marks the 50th anniversary of the creation of the biological macromolecular structures archive platform, known as the Protein Data Bank. Until now, 1198 unique membrane protein structures have been reported on this platform. In the next sections, we will focus our discussion on the analysis and experimental efforts carried out by various research groups for approximately 50 years, to attain complete, functional structures of the nAChR employing Cryo-EM and X-ray diffraction, as outlined on Table 1.

## 7. Methods Historically Used for nAChR Solubilization

A fundamental obstacle to achieving a high-resolution structure of nAChRs is the preparation of milligram amounts of stable, homogeneous, and functional nAChR detergent complexes (nAChR-DCs) for crystallization trials. Undoubtedly, key to the preparation of suitable nAChR-DCs for structural studies is preserving the stability and functionality of the nAChRs ion channel translocation machinery.

In muscle, the topology of nAChRs, their stability, lipid-dependent activity, and interaction with synaptic proteins create a complex scenario that hinders their pure isolation. In addition, since the nAChR is glycosylated at the extracellular loop, the N-linked glycosylation can significantly influence the conformation of nAChRs during solubilization and crystallization processes [67,68,69]. Typically, membrane protein solubilization is accomplished through the use of detergents. The physicochemical properties of these detergents need to be compatible with the annular lipid composition of the native protein, to minimize disruption of the protein’s hydrophobic belt, preventing premature delipidation, which could lead to conformational changes and aggregation [70,71,72,73].

Traditionally, conventional purification methods used to obtain high-purity nAChRs were not efficient, allowing high levels of impurities from ATPase, rapsyn, calcium channels, tyrosine kinase, and agrin (Table 2) to co-purify with the nAChR. However, different methods, such as high-resolution separation columns, in conjunction with emerging protein separation technologies, have allowed researchers to obtain preparations with a high level of purity. Different affinity chromatography methods, alkaline treatments, use of chaotropic salts, and sucrose gradient procedures have been commonly employed to purify nAChRs [74,75,76]. Initial nAChR purification attempts were made using chromatography columns with resins conjugated to different cobra toxins to capture the muscle-type nAChR isolated from the electric organ of the eel Electrophorus electricus (Ee) and the ray Tm. Although sufficient purified protein yields were obtained, nAChRs were not functional, thus demonstrating the need to develop novel nAChR purification protocols exerting minimal disruption to their native lipid environment.

Recently, our laboratory developed a novel, straightforward, and sequential automated purification process to obtain highly pure and functional nAChR-DCs [77,78]. First, the protein is solubilized using LysoFos Choline 16 (LFC-16), a phospholipid analog detergent, supplemented with CHS, and sodium chloride at high concentrations to assist the solubilization process, thus removing impurities that have weak or non-specific interactions with the nAChR. Overall, the purification process comprises the implementation of three sequential chromatography steps: the first uses a bromoacetylcholine affinity chromatography for the reverse capture of nAChR-DCs. The second chromatographic step involves the use of Capto lentil lectin chromatography, to ensure the exclusion of any protein or contaminant lacking glycans. The third step uses size-exclusion chromatography (SEC), which allows the separation of previously eluted glycan-containing proteins by their hydrodynamic volume. Furthermore, to address the nAChR stability issue, we supplemented samples with a buffer containing cholesterol during all chromatographic steps. Importantly, the first purification step was adapted to an automated medium pressure system for affinity protein capture. We also changed the previously used Affi-Gel 10 to Affi-Gel 15 resin, based on the isoelectric point of the protein under study. This resin was specifically conjugated under anhydrous conditions, packed, and loaded into GE XK columns designed to run medium to low-pressure chromatographic processes. Moreover, the column was placed within an insulated jacket using a recirculation system to control the temperature inside the column, in addition to performing experiments in a cold room. The combination of all of these steps and strategies has substantially improved the quality and functional parameters of purified nAChR-DCs as a possible final product for the production of high-quality crystals [77,78]. It has been postulated that the main impediment to achieving a high-resolution structure of the nAChR is the purity, homogeneity, function, and stability of the nAChR-DCs [52,54,55,73,79,80,81,82,83]. Therefore, protein purification is a fundamental first step for high-resolution structural studies. Procedures, protocols, and methods leading to production of high-yield, high-purity nAChR-DC, which retains its stability and function, enhance opportunities for harvesting diffractable crystals.

## 8. New Methods for nAChR Detergent Solubilization

Since Electrophorus and Torpedo membranes were discovered to be enriched with nAChRs, several research groups around the world studied different and/or similar strategies to obtain highly pure, functional nAChRs. However, few have focused on the importance of generating functional nAChR detergent complexes. The above-described purification protocols proved capable of producing a stable and functional nAChR-DC, suitable as a starting material for crystallization studies [77,78]. Among the most common methods used in these cases are sucrose density gradient, alkali treatment, ion exchange, gel filtration, and different affinity chromatography methodologies using nAChR ligands [62,84,85,86], some of which have the disadvantage of short lifespans one (1) month or less [87].

Since 1971, the majority of nAChR purification efforts have used Triton X-100 as a suitable solubilization detergent (Table 2). It is now known, however, that its usage produces a negative effect on acetylcholine binding, thus affecting ion channel function [74]. Thus, some have used β-octylglucopyranoside and cholate as alternatives. Our research group has evaluated the effect that detergents have on the stability and function of nAChR-DCs. Indeed, we assessed the solubilization process using detergents with physicochemical properties similar to the constituents of the cell membrane where nAChRs are embedded [52,54,73]. Specifically, we selected lipid-analog detergents whose biophysical, chemical, and biochemical properties resemble the natural environment of these integral membrane proteins in Tc electric organ [73].

We used lipid-analog detergents such as n-dodecylphosphocholine (FC-12), n-tetradecylphosphocholine (FC-14), n-hexadecylphosphocholine (FC-16), 1-dodecanoyl-sn-glycero-3-phosphocholine (LFC-12), 1-tetradecanoyl-sn-glycero-3-phosphocholine (LFC-14), 1-palmitoyl-2-hydroxy-sn-glycero-3-phosphocholine (LFC-16), (3-[(3-Cholamidopropyl) dimethylammonio]-1-propane sulfonate) (CHAPS), sodium cholate (3α,7α,12α-Trihydroxy-5β-cholan-24-oic acid)], and, N,N′-bis-(3-d-Gluconamidopropyl) cholamide (BigCHAPS). Additionally, we have used other commonly used detergents that have been used successfully in membrane protein structural studies, including 6-Cyclohexyl-1-hexyl-β-d-maltoside (Cymal-6), n-Dodecyl-β-d-maltopyranoside (DDM), lauryldimethylamine-N-oxide (LDAO), n-Octyl-β-d-glucopyranoside (OG), and polyoxyethylene-(9)-dodecyl ether (Anapoe-C12E9) (Table 2). Among these detergents, LFC-16 was the only one able to solubilize Tc nAChR-enriched membranes without producing significant native lipid depletion, activity, or destabilization [52,54,55,79]. The functionality of nAChR is highly dependent not only on phospholipid composition, but also on the number of cholesterol molecules that surround each subunit. It is well established that cholesterol modulates nAChR function [88,89,90,91]. As previously mentioned, our solubilization approach recognizes the importance of cholesterol and includes it as part of the procedure used to solubilize and isolate nAChR-DCs. We thereby provide greater stability to the protein during solubilization and purification, preventing the potential loss of cholesterol molecules via premature delipidation.

A close look at all of the articles reporting isolation and purification of muscle-type nAChRs from different species of Torpedo shows very low yields and recovery upon purification. In fact, the literature shows that 120–1000 g of Tc tissue yields between 2.3 and 50 mg of purified nAChR [91,92]. Another organism used to obtain nAChR was Electrophorus electricus (Ee), from which only 3–4 mg of purified nAChR was recovered from 500 g of tissue [93]. In our method, a significant reduction in the amount of Tc tissue used for solubilization and purification (20–40 g) is possible, from which we were able to produce 2–4 mg of pure, stable, and functional nAChRs [77,78]. As mentioned above, procedures that have been applied to the purification of nAChR from other tissue sources have invariably yielded nAChRs with several co-purifying contaminants.

## 9. Assessment of Purity and Stability of nAChR-DCs

We have developed and characterized a highly reproducible purification process for native nAChRs with final purity levels of ~94% [77,78]. We implemented, for the first time in native nAChR samples, a robust purity analysis process using microfluidics-based capillary gel electrophoresis using the Agilent 2100 Bioanalyzer system. Advantages of microfluidics-based automated electrophoresis over traditional gel electrophoresis include dramatically reduced sample size (4 µL for proteins) and reagent consumption, significantly faster analysis time, and less hands-on activities during sample preparation and data analysis. This approach allowed us to precisely detect impurity levels in our preparations using a fluorescent dye that covalently binds to the epsilon amino groups of lysines on the native nAChR [121,122]. As part of our biophysical characterization protocols, we recently also studied the effect of cholesteryl hemisuccinate (CHS) on the nAChR-LFC-16 complex, using lipidic cubic phase fluorescence recovery after photobleaching (LCP-FRAP) [77]. We evaluated nAChR stability by incorporating CHS in the protein–detergent complex during the multistep purification processes developed in our laboratory (see Section 8). For these studies, we incorporated CHS and methyl-β-cyclodextrin treatments, monitoring them for 30 days. Results showed that, in the presence of CHS, the fractional fluorescence recovery, mobile fraction, and diffusion coefficient decreased significantly starting from the first day of the study, as compared with the samples not treated with CHS. This suggests a decrease in the fluidity of the lipid matrices that prevent adequate incorporation into the LCP, to promote crystallogenesis of the nAChR-DCs. However, when methyl-β-cyclodextrin was incorporated into nAChR-LFC-16 + CHS in solution, significant differences were observed. Notably, we obtained an 84% increase in the mobile fraction compared to the samples not treated with CHS. This suggests a tendency to stabilize the sample due to the removal of the excess cholesterol present, thus obtaining a suitable candidate for future crystallographic studies. On the other hand, we not only observed the stability in lipid matrices of LCP, but also studied the conformational changes that the incorporation of CHS and methyl-β-cyclodextrin can bring to the native nAChR-LFC-16 complex. To this end, we performed thermal unfolding studies using a Jasco CD-1500 circular dichroism spectrometer, to assess the conformational changes in the nAChR-LFC-16 complexes. Our results suggest that the incorporation of 0.2 mM CHS increased the helix structures by 32.6%, and that this CHS supplement protects the complex up to a level of thermal denaturation of 55.04 °C at 222 nm, compared to the samples not treated with CHS or treated with CHS + methyl-β-cyclodextrin [77].

## 10. X-ray and Cryo-EM Structural Studies of α4β2 nAChRs

More than three decades since the discovery of neuronal-type nAChRs, the structure of the human α4β2 nAChR was first elucidated by X-ray crystallography at a 3.9 Å resolution (PDB entry 5KXI, Figure 2) [7]. To achieve crystallization in a vapor diffusion setup, a recombinant version of the receptor was developed in which the “BRIL” gene was inserted between the M3-M4 loop of both subunits, a method that has been successful in the crystallization of G protein-coupled receptors [123]. The α4β2 nAChR is composed of five subunits in a pseudo-symmetrical arrangement, first observed in the muscle-type nAChR by Unwin in 1985, using early cryo-EM technology [124]. Each of the subunits is composed of a C-terminal domain that contains three α-helices that span the cell membrane and a fourth amphipathic MX helix. The N-terminal has an α-helix followed by a large extracellular domain containing ten β-strands. The central pore of the ion-channel is defined by the transmembrane domain 2 (TM2) of each subunit and begins at a large extracellular space that follows a funnel shape that spans the membrane with a diameter of approximately 5.4 Å. The crystal structure of this receptor was solved in a desensitized, non-conducting conformation. Its subunits naturally combine to form stoichiometries with high (α4(2)β2(3)) and low (α4(3)β2(2)) affinities towards natural agonists like ACh and nicotine. This is attributed to the amount of favorable agonist binding sites that form in the interface between α4 and β2 subunits. The subunit arrangement of the existing crystal structure was in the high-affinity α–β–β–α–β forming the pentameric ring, which was selectively produced and isolated through elegant expression experiments using recombinant Baculovirus (BV) transductions [125].

A similar study yielded the 3.5 Å 3D structure of the low affinity α4(3)β2(2) nAChR, this time making use of Cryo-EM. These subunits are almost structurally identical, and to discriminate between them, monoclonal antibodies were produced, and the high-affinity antigen-binding (Fab) fragments were used to identify β subunits. Insight into α–β interfaces revealed structural limitations that hinder the formation of other stoichiometries [48]. To explain why the only two stoichiometries that assemble by free energy were α4(2)β2(3) and α4(3)β2(2), homopentameric assemblies of each subunit (α–α or β–β were superimposed. The less favorable of the subunit combinations was from the β–β interface. The boundary between the two β subunits seems to create a polar aperture that disrupts the appropriate formation of the ion channel, making the predicted β-homopentamer unable to close the pentameric ring. In the case of α–α interfaces, the α4 subunits tend to have atomic clashes and higher surface area buried, meaning that the α-homopentamer is also unfavorable because it packs too tightly.

## 11. The Nicotine Binding Site of the α4β2 nAChR

Since the late 1980s, photoaffinity labeling experiments on muscle-type nAChRs from Tc have helped identify specific amino acids involved in the receptor’s binding site, which together with the earliest 3D structure, suggested that nicotine binding occurred in the interface of subunits (α, β, and γ) [126]. Because the identified amino acids were all aromatic, it was proposed that the binding of ACh was mediated through cation-π interactions, where an aromatic ring creates a region of negative electrostatic potential that allows the binding of ions with significant strength [127]. This interaction is always observed in neuronal nAChRs between one of the aromatics and agonists such as nicotine, but it does not occur in the muscle type nAChR [128]. This phenomenon contributed to the explanation of why nicotine consumption generates a physiological effect solely through high-affinity neuronal nAChRs and not the low-affinity muscle-type nAChR.

The neuronal subtype of nAChRs found in the central nervous system is associated with nicotine addiction, and exhibits the highest affinity towards ACh and nicotine [129]. This high sensitivity is caused by the number of favorable interfaces between α4 and β2 subunits in the α4(2)β2(3) stoichiometry, where each subunit contributes a specific aromatic residue, creating the binding site or “aromatic box”. This early understanding of nAChR binding sites, without a high-resolution 3D structure, was advanced through pioneering studies on the structure of the soluble AChBP from the snail Lymnaea stagnalis [24]. AChBP is released from glial cells, and acts by modulating cholinergic synaptic transmission by binding excess ACh in the synaptic cleft. In its natural form, the AChBP forms soluble pentamers that are homologous to the N-terminal domain of pentameric ligand-gated ion channels. The hydrophilic nature of AChBP and its ability to bind nAChR natural agonists made it the ideal model for high-resolution structural studies using X-ray diffraction. The resulting AChBP 3D structures confirmed all the previous mutational and biochemical studies on nAChRs binding sites [130] (Figure 2).

Nicotine binds selectively in the α–β interface of the neuronal α4β2 nAChR, and has minimal contact with the solvent. The binding pocket can be divided into two sides formed by the subunits: the α4 forms the (+) side of the pocket; the β2 forms the (−) side. The “aromatic box” is formed by residues Thr-100 from loop A and Trp-57 in loop D belonging to the β2 strand, while the back walls are composed of residues Trp-156 and Leu-121 of the β6 strand in loop E. Loop C forms the front wall of the pocket, and firmly packs the ligand together with interactions from nearby cysteines, and the residues Tyr-197 and Tyr-204. The top of the binding pocket is formed by Val-111 and Phe-119 in loop E. In agreement with studies on the nature of ligand binding, nicotine was shown to naturally form a hydrogen bond between its pyrrolidine nitrogen and the carbonyl backbone of the Trp-156 to create the cation-π interaction commonly present in this receptor superfamily.

## 12. Sterol Binding Sites of the α4β2 nAChR

The recent neuronal α4β2 nAChR 3D structures have provided the most structurally detailed insight into their architecture, stoichiometric assembly, nicotine binding sites, and protein–lipid interactions at the annulus of the receptor’s transmembrane domain (TMD). The latter, however, has widely been elusive in structural studies, since tightly bound native lipids were typically removed during detergent solubilization processes. Another limitation is that previous cryo-EM electron density maps were not able to provide enough resolution at the receptor’s annulus [131], which made analysis of protein–lipid interactions inaccessible from a structural point of view [132,133]. The initial electron density observed at the periphery of the TMD was in the prokaryotic pentameric ligand-gated ion channel high-resolution crystal structures, from which two of the three lipid-binding sites were identified for each subunit [134,135]. Recent advances in the analysis of lipid nanodisc-embedded proteins using single-particle Cryo-EM, together with nanometer-sized membrane fragmentation using amphipathic copolymers, has provided an adequate tool to study the structure and function of these protein–lipid interactions [136,137]. This combination of methods recently yielded the most detailed model of protein–lipid interactions at the TMD of pentameric ligand-gated ion channels and identified two different phospholipid molecules bound to each subunit at atomic resolution [138]. In the case of the reported neuronal nAChR structures (α4β2, α7, and α3β4), no electron density has been detected for annular phospholipids, mainly because all protein preparations involve detergent solubilization, in contrast to the native membrane environment provided by styrene-maleic acid nanodiscs in Erwinia ligand-gated ion channel studies. The only existing X-ray crystal structure of a neuronal nAChR is the human α4β2 nAChR and no electron density belonging to bound lipids was detected in the TMD of any subunit. However, similar studies using single-particle Cryo-EM reported “sausage-shaped” electron densities between subunit interfaces interacting with both the subunit and an adjacent cholesterol molecule in a 3.5 Å structure of the α4β2 nAChR [48,49] (Figure 3). The orientation of bound cholesterol seems to be dictated by what the authors label the “principal (+)” side of the subunit interfaces, which differs from binding sites for cholesterol and potentiating and inhibitory neurosteroids in GABA_A_ receptor chimeras [139,140]. In the human α4β2 nAChR (PDB entries 6CNJ and 6CNK, Figure 2), cholesterol binds to a concave surface in a junction formed by the MX, M3, and M4 helices. The only variation in the interface between α4 and β2 subunits is a unique amino acid penultimate helical turn of the M3 transmembrane helix that shows different chemical properties and dictates whether cholesterol binds to the apical portion of the binding regions. In the α4 (+) subunit, several amino acids form the cholesterol-binding site. The cholesterol molecule located at the M4 interacts with amino acids Phe-361, Ile-357, Trp-350, and Phe-300; in the M3, interactions are with Leu-293 and Phe-300 and Arg-307. This cholesterol molecule also interacts with the M1 of the β2 (−) subunit (Ser-225, Ile-28and Tyr-355) and with the M4 (Leu-354, Trp-355 and Arg-351), Figure 3. While this cholesterol binding site is consistent with photo-activable cholesterol analog mapping studies [141], there were no electron densities detected belonging to bound phospholipids. It seems evident that emerging Cryo-EM technology, together with detergent-free methods, provides an appropriate experimental setup to study, in high resolution, the protein–lipid interactions of membrane proteins in their native lipid environment, specifically nAChRs.

## 13. The Cryo-EM Structure of the α3β4 nAChR

Another recent Cryo-EM structure of related nAChR, the α3β4, has been elucidated by Ryan Hibbs’s laboratory [49]. The α3β4 modulates the mesolimbic dopamine system, and has been implicated in drug addiction [142,143]. This time, Hibbs’ group added new tools to their crystallization arsenal, by using Fabs against the α subunit and replacing the intracellular portion of the M3-M4 loop with the soluble BRIL protein, reducing the predicted disorder in this region. In addition, the α and β subunits were labeled with green fluorescent and mCherry proteins for determination of the relative stoichiometry of each subunit under different expression conditions. Additionally, to attain a more suitable lipid environment, the α3β4 nAChR was reconstituted into lipid nanodiscs using saposin, soy lipids, and CHS. With this approach, they obtained four different structures from a single particle of the α3β4-Fabcomplex in nanodiscs using Cryo-EM: (1) α3β4 with nicotine/DDM at 3.34 Å (PDB entry 6PV7), (2) α3β4 with the high-affinity ligand AT-1001/DDM at 3.87 Å (PDB entry 6PV8), (3) α3β4 with nicotine/without CHS at 4.70 Å, and (4) α3β4/AT-1001/nanodiscs at 4.58 Å. The overall architecture looks like a cone, with a counterclockwise arrangement of subunits of α, β, β, α, and β, with the N-terminal domain in a distinctive β-sandwich, followed by the transmembrane domain embedded in the nanodiscs. Additionally, nicotine and water densities were observed at the binding pocket, and similar to the α4β2 structure, CHS was present at the edge of the transmembrane domain. Notably, the BRIL density was absent in all electron density maps.

Comparison and superposition of the amino acids that make up the nicotine binding pocket in both α4β2 and α3β4 reveal that all but two residues have the same orientation and identity [7,48]. The most relevant difference came from a substitution in β of Phe by Ile at the E loop, the Phe residue side-chain suggests stabilization of the nicotine pyridine ring by π-π interaction [144]. In addition, there is a slight outward displacement of loop C compared to the same region in the α4β2 nAChR structure, the displaced loop C produced a less tight binding pocket in the α3β4 nAChR. This could shed light in molecular terms on the low affinity of the α3β4 for acetylcholine, nicotine, and epibatidine compared to its relative α4β2 [145,146,147].

There is substantial evidence from binding studies and electrophysiological data regarding the difference in affinity of the partial agonist AT-1000 towards α3β4 and α4β2 nAChRs, despite the high degree of homology in the residues involved at the postulated binding pocket [7]. The α3β4 nAChR presents an approximately 3 to 200-fold greater selectivity for AT-1001 in rats and humans, respectively, versus the α4β2 nAChR [47,148]. The model from Cryo-EM density map from α3β4 ligand AT-1001/DDM complex presents sufficient information about the location of the large bicyclic ring and phenyl at the binding pocket. Following the line of thought outlined above, the α3β4 nAChR features an enlarged binding site compared to the α4β2 nAChR, which allows it to accommodate AT-1001 preferentially. Surprisingly, the AT-1001 adopted different orientations at the two binding sites. The author attributed the difference in orientation to a steric interaction between the phenyl bromine atom on AT-1001, with the side chain of Phe-119 on β subunits in both sides [48,49]. However, for a clearer picture of this type of interaction, X-ray crystallographic data at high resolution will be necessary, especially since the bromine atom electron density was unrepresented in the reported data.

The lipid dependence of the nAChRs has been widely reported [149,150,151,152]. To date, the crystallographic structures and Cryo-EM obtained present receptors in the desensitized state [7,48]. To evaluate the effect of lipids, the structure of α3β4 bound to AT-1001 in the presence of DDM was solved in nanodiscs at 4.58 Å resolution [49]. Apparently, the overall structural architecture of the complex resembles the previous structures in the desensitized state. The strategy implemented in the construction of a less disordered intracellular domain in the structure of the α3β4 nAChR allowed examination of some features of the short post-M3 helix (MX), which end with the membrane-associated helices until the M4, and form the end of the receptor channel [153]. The cytoplasmic loop domain contains important amino acids and secondary structure features that regulate trafficking [154,155]. These intracellular domain helices are similar in position in all subunits, with some differences in the localization of the charged residues depending on the nAChR subtype. It has been shown that the cytoplasmic residues of membrane-associated helices in the 5-hydroxytryptamine (5-HT3) receptors or serotonin receptors, a member of the Cys-loop family of ligand-gated ion channels, in addition to being part of the intracellular portals, also contribute to ionic selectivity [156,157]. The model obtained for the α3β4 nAChR showed the orientation of amino acids in the membrane-associated helices, in particular, those with negatively charged side chains, especially the Glu-432 and Asp-432 at the α subunit and Asp-420, Asp-421, Asp-443, and Glu-428 at the β subunit. The 3D orientation of these amino acids in the portal and the disposition of MX helix along with the membrane-cytoplasm interface, compared with 5-HT3 structures, suggest that ions permeate through the portals [156,158,159].

## 14. The Cryo-EM Structure of the Neuronal α7 nAChR

The Cryo-EM structure of α7 nAChR has recently been obtained in three different channel conformation states [64]. First, the α7 nAChR ligated to antagonist α-Btgx produced a resting-like closed-channel state with a density map at 3.0 Å resolution (PDB entry 7KOO) (Figure 2). Second, the positive allosteric modulator PNU-120596 and agonist epibatidine/α7 nAChR complex produced a stable, open channel state with a resolution of 2.7 Å (PDB entry 7KOX). Third, a desensitized closed channel state produced by epibatidine/α7 nAChR complex with a resolution of 3.6 Å (PDB entry 7KOX). At first glance, the three structures present a general architecture similar to the other receptors previously discussed, including the 5-HT3. The α7 nAChR structure mapping in the resting channel state was achieved using α-Bgtx, isolated from the venom of Bungarus multicinctus. α-Bgtx is a member of the crescent family of snake toxins that contain three distinctive finger-like loops (3FTx) that run parallel from the protein core, and are classified as a short or long peptide, 60 and 70 amino acids, and 4–5 disulfide bonds, respectively. Finger II of α-Bgtx shows the highest interaction in the neurotransmitter binding pocket, residues Phe-32 and Arg-36 located at the tip of finger II form a π-cation stack with the α7 nAChR Tyr-187 at C loop. These residues have been identified previously in the crystal structure of the complex between α-Bgtx and the α7 nAChR/AChBP chimera [160]. In addition, the binding interaction is also stabilized by several hydrogen bonds between α-Bgtx Gln-71 and receptor Lys-191, Glu-192, and Tyr-194. Other residues at the tip of the finger I of α-Bgtx, that have been involved in direct interaction with the loop C of α7/AChBP chimera, such as Pro-10 and Ile-11, do not appear to have the appropriate distance or orientation to carry out these interactions in the structure of α7 nAChR [160,161].

The α7 nAChR displays three states: (1) resting-like closed-channel, (2) stable open-channel state, and (3) desensitized closed-channel state. These present clear differences in conformation, along with their topography. The top view looking down the central axis presents the resting state enabled by α-Bgtx, with expanded star geometry and a pore that is smaller in diameter than the activated state [162]. Epibatidine binding at the subunit interfaces induces a concerted conformational change through the entire receptor. In the extracellular domain, the loop C of each subunit adopts a more closed conformation and the β sandwich is twisted. When comparing the map of the states induced in α7 nAChR by aligning the extracellular region, a displacement or tilt of the M1, M3, and M4 helices with respect to the normal of the bilayer was observed. The aforementioned transmembrane region showed a tilt of 8° degrees in the resting state and 25° in the activated state and then relaxed to 15° in the desensitized state. The M2, which is associated with the M4 via a strong salt bridge between M2-Lys-238 and M4-Asp-445, is pulled outward during activation [163]. The conformational changes at the transmembrane domain also affected the channel diameter that is flanked by the M2 pore-lining residues. The minimum distance between residues in the resting and desensitized state in the α7 nAChR was calculated to be 2.4 Å and 4.3 Å, while the activated state of minimum distance at the middle of the permeation pathway was 7.2 Å. The diameter of the latter is 1 Å shorter than the mouse nAChR pore which presented the narrowest part of the pore at 8.4 Å, but in the range of pore diameter for cation channel transporter [164,165]. The conformational changes that occurred at the extracellular domain are also translated into the intracellular region. Interestingly, the α7 nAChR Cryo-EM map revealed a short C-terminal amphipathic helix at the end of the M4 that breaks making a 90° turn that runs parallel to the membrane. This section was called the latch and has a role in channel opening.

## 15. The Most Recent Cryo-EM Structure of the *Tc* nAChR

The most recent structure of the native muscle-type nAChR was achieved through Cryo-EM of an nAChR-α-Bgtx complex preparation isolated and purified from Torpedo californica at a 2.69 Å resolution [65]. The nAChR was solubilized with Triton X-100 and exchanged into DDM to restore functionality, followed by reconstitution with saposin and soy polar lipids [49,117]. Proteoliposome patch-clamp experiments confirmed the functionality of this receptor preparation by carbachol stimulation, also, antagonist α-Bgtx blocked this activity. The model of the Tc nAChR-α-Bgtx complex presents an architecture similar to that obtained for the nAChR isolated from the Tm electric organ at 4 Å resolution (PDB entry 1OED and 2BG9) (Figure 2) [60,131]. It shows an almost pentameric symmetrical subunit arrangement with a clockwise order seen from the normal bilayer of (α, β, δ, α, γ). The two α-Bgtx molecules are to some degree embedded into the α-δ and α-γ interfaces in the extracellular domain. The γ and δ subunits present a noticeably extended F loop at the extracellular domain, all subunits present glycosylation sites. Most of the α-Bgtx molecule stabilizing interactions are facilitated by alpha subunits loop C and an N-glycan branch from the Cys-loop and complementary interactions from Loop F from δ and γ subunits. The α-Bgtx finger II appears to internalize below the C loop, and penetrates into the acetylcholine binding pocket in a manner that precludes acetylcholine molecule binding. Finger II of α-Bgtx in the α7 nAChR model penetrates more deeply into the binding pocket than in Tc due to a smaller Loop F in α7 nAChR [64]. The functional property involved in Loop F in the complementary site of interaction with α-Bgtx could be the cause of the differences in affinity and selectivity of nAChRs for 3TFx, since this loop is poorly conserved in the Cys-loop super family [166,167]. The residues Phe-32 and Arg-36 at the small helix-like tip of α-Bgtx finger II are parallel, oriented into the acetylcholine binding pocket. The residues Tyr-93, Tyr-190, and Tyr-198 at the α subunit C loop flank and the guanidinium group of the Arg-36 in the α-Bgtx finger II forms a π-cation sandwich. Additional stabilization comes from a hydrophobic cave between the Phe-32 of the toxin and Trp-149 of the alpha subunit and γTrp-55 or δTrp-57, depending on the type of interface (α-δ and α-γ).

Due to alleged discrepancies and registration inconsistencies in the first structures obtained from the nAChR from Torpedo marmorata, the authors were unable to make comparisons with its structure on the transmembrane domain region. However, when comparing the truncated previous structures obtained from tubular crystals of nAChR (PDB entry 2BG9) from Torpedo marmorata at 4 Å with the structure at 2.7 Å (PDB entry 6UWZ) (Figure 2), it was found that the former presents a much wider upper pore. Nevertheless, comparison between the protein–lipid architecture of the nAChR in tubular vesicles from the Torpedo marmorata electric organ and a more recent structure at 5.8 Å with the nAChR structure at 2.7 Å presents new facts to be considered [66]. The nAChR protein–lipid architecture at 5.8 Å confirms the presence of cholesterol microdomains and densities that suggest substantial interaction of cholesterol with receptor M1, M3, M4, at both bilayer leaflets and post-M3 helix (MX). The presence of CRAC motifs has been reported in the borderline of the extracellular domains and the TM domain in subunits α, β, and δ and CARC motif at the α and β subunits [168,169].

The densities in the inner leaflet of the bilayer for both models of solubilized nAChR-α-Bgtx complex (PDB 6UWZ) and the nAChR-vesicle matches quite well, but deviates substantially from the densities in the outer leaflet. These discrepancies are due to the inter-helical spacings between M4-M1 and M1-M3 that in the presence of cholesterol have a value of 14.2 Å and 12.8 Å, respectively, vs. approximately 2 Å closer in the nAChR-α-Bgtx complex structure. In the inner leaflet, the M3 and M4 are closer to each other, and cholesterol molecules are more likely to interact at the transmembrane lipid exposed face, such as in the α4β2 nAChR structure [48]. Both experimental and in silico data suggest the presence of internal, as well as peripherical sites in the nAChR capable of containing cholesterol molecules and lipids, whose presence stabilizes the protein structure [169,170,171]. Both structures show the intracellular domain formed by an amphipathic MX, a disordered linker, and an MA helix that precedes and continues towards M4 [65,66].

The amino acid residues at the MX-helix of nAChR subunits have been implicated in molecular signals that regulate the assembly, trafficking, and expression of muscle nAChR [172,173]. Furthermore, the crystal structures of the human neuronal α4β2 and the cryo-map densities of α7 nAChR, Tc, and Tm have shown the presence of this secondary structure. Although there is no evidence of a cholesterol-sensitive motif in the MX amino acid sequence, cholesterol microdomains have been observed in the vicinity of the MX helix in tube-shaped vesicles of the nAChR from Tm [66]. The amino acid composition and orientation of the MX vary in the nAChR family; however, each of them contains enough hydrophobic and polar amino acids for interaction between the lipid–water interface of a membrane [48,64,65,66]. Probably, part of the MX penetrates the water–lipid interface by disrupting the phospholipid-to-phospholipid head group interaction, consequently producing some kind of lipid disorder that accommodates cholesterol molecules. So far, a substantial advancement has been achieved in the number of nAChR structures resulting from electron density maps of several nAChR subtypes, mostly worked in Ryan Hibbs’ laboratory. This advancement is largely due to the development of molecular strategies to stabilize the intracellular region by partial deletions, insertion of glutamate–arginine linkers, and selective fluorescence labeling of nAChR subunits and monoclonal antibodies, all of them developed by Hibbs and collaborators. In addition to technological advances in the area of electron microscopy, hardware development, sample preparation, new software data acquisition, and modeling of electron density maps. However, it is still necessary to obtain whole structures of the nAChRs in their native environments to preserve the inherent conformation, stability, and functionality of these important class of membrane proteins.

## 16. Summary of nAChR Structures: Lessons Learned

At the present time, there are 22 EMDs, 21 PDBs, and 3 PDBs not yet reported for structures (α3β4 and muscle-type Tm) of nAChRs solved since 2013 (Table 1). The only X-ray structure so far is the α4β2 nAChR (PDB 5KXI) Figure 2, with a resolution of 3.97 Å. The range of structure resolutions is 50 Å (PDB 4BOG) to 2.69 Å (PDB 6UWZ). The best resolution so far has been obtained for the muscle-type Tm nAChR using Cryo-EM. Most of the reported structures are from Cryo-EM reconstructions from tomography and single particles in the activated or closed channel state. The recent Cryo-EM α7 nAChR structures have provided detailed structural information of resting, activated, closed, and desensitized states. Molecular mechanic simulations of these α7 nAChR structures were used to elaborate a structural gating mechanism for these subtypes. The most relevant structural features emerging from the recent nAChR structures are:(1)Agonist and antagonist binding sites. The agonist binding site of nAChR is located and embodied within six canonical loops in the ECDs, in the interface between a primary subunit (loops A–C) and a complementary subunit (loops D–F). The emerging structures of the nAChRs have confirmed that the signature loop C, which undergoes a marked conformational change upon agonist binding, contains the agonist binding site. The α4β2 nAChR X-ray structure and the Cryo-EM structures for α4β2, α3β4, α7 and muscle-type Tm nAChRs confirm previous biophysical studies for the location of amino acid residues and glycans involved in agonist binding site. In particular, the structure of the muscle-type Tm nAChRs-Bgtx complex at 2.7 Å resolution confirms side-chain contacts in the orthosteric site (αTyr-93, αTyr-190, αTyr-198), and stacks in a cation-π sandwich between α Tyr-198 and Phe-32. Additionally, αTrp-149, γTrp-55, δTrp-57 participate in ligand binding. In addition, this structure shows a cation-π sandwich that corresponds to αTyr-198-Bgtx-R36-Bgtx-Phe-32, which presumably indicates the pivotal binding site for neurotoxins to nAChRs including α7 nAChR.(2)Structural basis for differences in agonist affinities. The Cryo-EM structure of the α4β2 nAChR uncovers two agonist binding sites, α4–β2 and β2–α4. The α4–β2 has the highest affinity for nicotine. Superposition of the α4β2 and α3β4 binding sites revealed that the side chains in contact with nicotine and their orientations are conserved. An outward shift in loop C upon nicotine binding results in a less compact agonist binding pocket of the α3β4 nAChR compared to α4β2 nAChR, which could explain the reduced affinity for nicotine in the α3β4 nAChR. Additionally, the less conserved loop E from the complementary face of the agonist binding site contains side chains that are close to the agonist could contribute to affinity ranges.(3)Two stoichiometries of the α4β2 nAChR. Cryo-EM data confirmed two stoichiometries (α4(2)β2(3)) and (α4(3)β2(2)) previously suggested by functional studies [174].(4)Permeation pathways. The structure of the muscle-type Tm nAChRs-Bgtx complex provides the best atomic resolution for the nAChR ion permeation pathway in the closed state. The permeation pathway consists of an electronegative extracellular vestibule and tightly closed ion channel pore with a polar intracellular domain. The M2 helices from each subunit have two hydrophobic constriction points, one at the extracellular side of the pore, at the 16′ and 9′ positions. These two gates at 16′ and 9′ positions are the main obstructions for hydrated ion permeation.(5)Gating mechanism. The Cryo-EM maps of the α7 nAChR bound to α-Bgtx, epibatidine + PNU-120596, and epibatidine alone in combination with an elegant electrophysiological characterization of the EM construct expressed in HEK293S GnTl- cells in the absence and presence ligands and prior to solubilization of the α7 nAChR with DDM, has provided the structural basis for the resting, activated and desensitized states. Molecular dynamics simulations were used to predict a model for the α7 nAChR gating mechanism.(6)The role of the latch turns in the α7 nAChR gating. Interestingly, the α7 nAChR structures revealed residues Pro-469 and Ala-467 in the latch turn are essential for coupling agonist binding and channel gating. The mechanism underlying the contribution of the latch turn on the nAChR ion channel gating remains to be elucidated.(7)Structural basis for the high calcium permeability of the α7 nAChR. A glutamate side-chain located in position 97 (Asp-97) in the extracellular vestibule that forms a salt bridge with residue K124 in the resting state is pivotal for the high calcium permeability. A 6.4 Å constriction of the ECD during agonist activation removes the salt bridge and reorients E97 into the center axis of the permeation pathway to secure calcium conductance. Interestingly, in the desensitized conformation the salt bridge is reestablished.(8)The role of transmembrane (M1, M3, and M4). Molecular dynamics simulations of the α7 nAChR structures suggest that the M1, M3, and M4 tilts during channel activation from 8 degrees (resting state) to 25 degrees (open channel state) and then relaxes to 15 degrees in the desensitized state. During channel activation, M4 slides up four residues, consistent with functional studies that proposed a spring model for the M4 by Otero-Cruz et al., 2007 [175]. The slide of the M4 lipid-exposed helix results in a shift of the M2 helix away from the pore axis resulting in channel opening. The role of M4 in channel gating have been previously described by structure-function studies (http://nachrs.org/ (accessed on 21 September 2021)). Along these lines, the upward shift of the M4 helix drags the MA TMD resulting in a subtle uncoiling of the M3-M4 loop which has may have consequences in channel activation and desensitization.(9)Structure of intracellular domains (ICDs). Structural information of the ICDs of the nAChRs has remained ambiguous, due to the need to delete a large portion of this domain and or modification with YFP or the thermostable BRIL protein. Nonetheless, the muscle-type Tm nAChRs-α-Bgtx complex provides structural information of a “native ICD” in the closed state. This structure shows a “hydrophobic plug” of EM density surrounded by hydrophobic residues, similar to the modified ICD of the α3β4 nAChRs. Molecular dynamics simulation of the α3β4 nAChR suggested that a lipid molecule is needed to stabilize the non-native ICD. The five MA helices of the muscle-type Tm nAChRs-Bgtx complex converge in a narrow constriction (~5.4 Å) at the cytosolic end, which permits mobilization of hydrated sodium ions. The five portals located at the interface of the MA helices in the α3β4 nAChRs contain fenestrations encompassed by acidic and polar side chains, providing the required architecture for the cation exit pathway.

## 17. nAChR Challenges and Future Perspectives

During the past four years, substantial advances have been achieved in the number of nAChR structures, mostly contributed by the Ryan Hibbs laboratory. These breakthroughs are largely due to the development of molecular strategies to stabilize the ICDs by substantial deletions, removal of glycans, insertion of a glutamate-arginine linker and thermostable bRIL protein, selective fluorescence labeling of nAChR subunits, and the use of monoclonal antibodies, developed by Hibbs and collaborators. Most importantly, the technological advances in the area of Cryo-EM microscopy, hardware development, sample preparation, new software data acquisition, and modeling of electron density maps have been fundamental to the advances in nAChR structural biology. Despite all the aforementioned technological advances that have led to four (4) novel structures during the last four years, there are some limitations in these structures due to the intrinsic nature of their constructs and technical challenges that must be undertaken with special attention: (1) atomic resolution of the nAChRs must be enhanced, (2) structures of open and resting conformations are still needed for some of these receptors to attain more detailed information on the gating mechanisms, (3) high-resolution structures of neuronal nAChRs complexes with agonists, antagonist, and different allosteric modulators are highly important for drug discovery, (4) high-resolution structures of the unmodified ICDs, in particular, the M3-M4 domain of neuronal nAChRs, remain a challenge but are highly needed to understand the structural and functional basis of this domain, and (5) structures of other neuronal nAChRs subunits are granted. To overcome these challenges, it will be important to improve the quality and stability of the nAChR detergent complex. Continuous improvements in purification methods (outlined in Table 2) and new technologies to assess purity [77], stability, and functionality will serve to enhance the structural biology of the nAChRs. Moreover, it is important to continue efforts to obtain native nAChR structures, which remains the biggest challenge in the field of structural biology. As previously mentioned, the amount of data available on nAChRs is remarkably rich, and the physiological and clinical relevance of these ion-channel receptors has become highly valuable to understanding multiple diseases affecting humans. These recent nAChR structures will certainly define a roadmap for new discoveries, hopefully leading to the development of novel pharmacological ligands for the treatment of numerous neurodegenerative and neurological diseases.

## Figures and Tables

**Figure 1 molecules-26-05753-f001:**
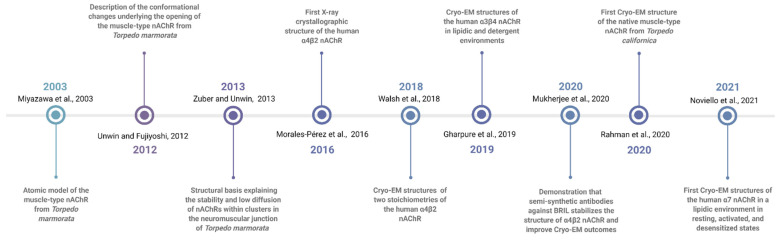
Most important milestones in the effort to obtain nAChR structures during the past two decades. Created with BioRender.com (accessed on 21 September 2021).

**Figure 2 molecules-26-05753-f002:**
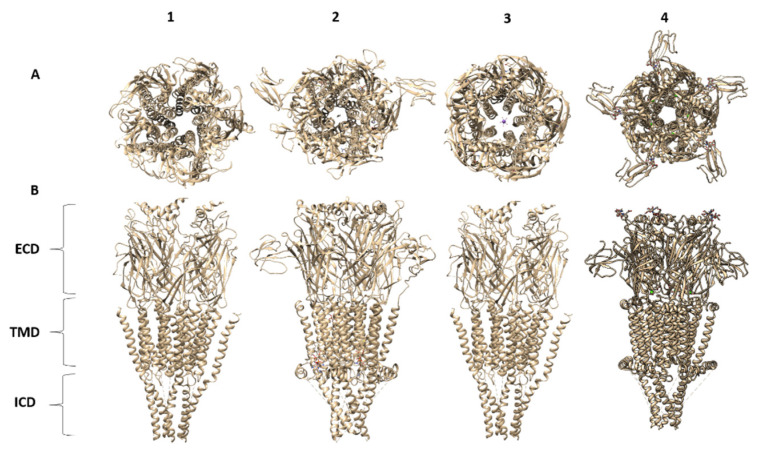
The most relevant structures of the pentameric nAChRs. (**A**) From 1 to 6, top view down the axis of a single nAChR. (**B**) From 1 to 6, vertical view from the side, perpendicular to the imaginary bilayer plane, showing the extracellular domain (ECD), transmembrane domain (TMD), and intracellular domains (ICD). (1) Cryo-EM of nAChR from Torpedo marmorata (PDB 2BG9), (2) Cryo-EM of nAChR from Torpedo californica linked to α-Bgtx (PDB 6UWZ), (3) X-ray crystallographic structure of the human α4β2 nAChR (PDB 5KXI), (4) Cryo-EM of α7 nAChR (PDB 7KOO), and (5 and 6) Cryo-EM α4β2 nAChR-nicotine-Fab complexes; 2α:3β (PDB-6CNJ); 3α:2β (PDB-6CNK). Molecular graphics and analyses were performed using Chimera from University of California San Francisco (UCSF).

**Figure 3 molecules-26-05753-f003:**
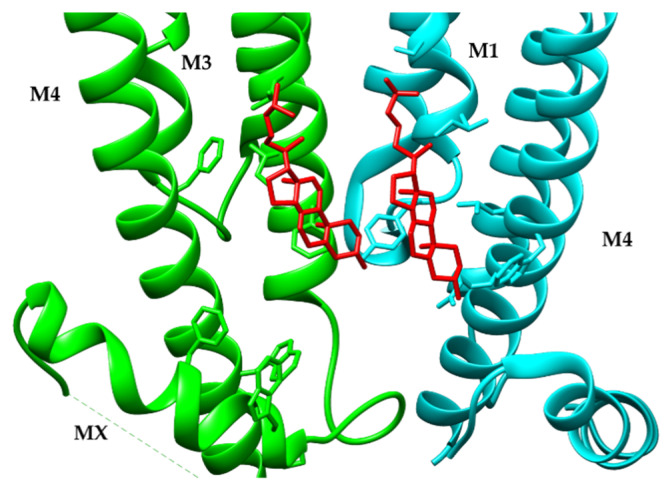
Structural Model for the cholesterol binding site in the α4β2-nicotine-Fab complexes. Using UCSF Chimera, an alignment of the α4β2 receptor-nicotine-Fab complexes was constructed for the 2α: 3β (PDB-6CNJ). Cholesterol (red) interacts with amino acid residues in the TM3 and TM4 transmembrane regions, the MX helix of one of the subunits (green), and the TM1 and TM4 (cyan) in a α-β interface.

**Table 1 molecules-26-05753-t001:** Structures of nAChR reported since 2003.

^1^ EMD	Protein Complex	Aggregation State	Reconstruction Method	Deposit Date	Resolution (Å)	Associated PDB	Organism	Ref.
1044	Ion channel nAChR	Filament	Helical	2003-04-03	4.00	1OED	*Tm*	[60]
1044	Ion channel nAChR	Helical array	Helical	2004-12-17	4.00	2BG9	*Tm*	[60]
2071	Ion channel nAChR	Particle	Single particle	2012-04-12	6.2	4AQ5	*Tm*	[61]
2072	Ion channel nAChR	Helical array	Helical	2012-04-12	6.2	4AQ9	*Tm*	[61]
2376	Ion channel nAChR/rapsyn	Particle	Tomography	2013-05-19	50.0	4BOG	*Tm*	[62]
2377	Ion channel nAChR/rapsyn	Particle	Tomography	2013-05-19	41.0	4BOI	*Tm*	[62]
2378	Ion channel nAChR/rapsyn	Tissue	Tomography	2013-05-19	41.0	4BON	*Tm*	[62]
2381	Ion channel nAChR/rapsyn	Particle	Tomography	2013-05-19	42.0	4BOO	*Tm*	[62]
2382	Ion channel nAChR/Rapsyn	Tissue	Tomography	2013-05-19	42	4BOR	*Tm*	[62]
2383	Ion channel nAChR/Rapsyn	Particle	Tomography	2013-05-19	42.0	4BOT	*Tm*	[62]
n/a	α4β2	Crystal	X-Ray diffraction	2016-7-20	3.94	5KXI	*Hs*	[7]
7535	2α:3β/nicotine of α4β2	Particle	Single particle	2018-03-08	3.70	6CNJ	*Hs, Mm*	[48]
7536	3α:2β2/nicotine of α4β2	Particle	Single particle	2018-03-08	3.90	6CNK	*Hs, Mm*	[48]
20487	α3β4/nicotine	Particle	Single particle	2019-07-19	3.34	6PV7	*Hs, Mm, Ec O11*	[49]
20488	α3β4 ligand AT-1001/DDM	Particle	Single particle	2019-07-19	3.87	6PV8	*Hs, Ec O11*	[49]
20489	α3β4/nicotine without CHS	Particle	Single particle	2019-07-19	4.70	n/a	*Hs, Ec O11*	[49]
20490	α3β4/AT-1001/nanodiscs	Particle	Single particle	2019-07-19	4.58	n/a	*Hs, Ec O11*	[49]
20857	α4β2/varenicline complex	Particle	Single particle	2019-10-22	3.71	6UR8	*HS, Ec*	[63]
20863	α4β2/varenicline complex/antibril	Particle	Single particle	2019-10-26	3.87	6USF	*Hs, Ec*	[63]
22983	α7 nAChR/epibatidine and PNU-120596	Particle	Single particle	2021-03-17	2.70	7KOX	*Hs, Ec*	[64]
22979	α7 nAChR/α-Bgtx	Particle	Single particle	2021-03-17	3.00	7KOO	*Hs, Ec*	[64]
22980	α7 nAChR/epibatidine	Particle	Single particle	2021-03-17	3.60	7KOQ	*Hs, Ec*	[64]
20928	Native muscle-type nAChR/α-Bgtx	Particle	Single particle	2019-11-06	2.69	6UWZ	*Tc*	[65]
11239	Ion channel nAChR	Filament	Helical	2020-06-29	5.8	n/a	*Tm*	[66]

Abbreviations: ^1^ EMD = electron microscopy data bank, *Tm* = *Torpedo marmorata*, *Hs* = *Homo sapiens*, *Ec* = *Escherichia coli*, *Ec* O11 = *Escherichia coli* O11, *Tc* = *Torpedo californica*, *Mm* = *Mus musculus*, CHS = cholesteryl hemisuccinate, DDM = n-dodecyl-β-d-maltopyranoside, Ref. = reference, n/a = not available. Note: The authors wish to acknowledge three key seminal studies not included in the table [57,58,59]. These studies were carried out using protein fibers but did not generate PDB or EMD structures.

**Table 2 molecules-26-05753-t002:** Purification strategies employed to produce highly pure nAChRs.

Tissue Source	Quantity of Tissue	Purification Method	Detergent(s)/Solvent	Reported Yield	Ref.
^1^ *Tm*	n/a	Sephadex G-200/sucrose density gradient	Triton X-100	n/a	[94]
*Tm*	60 g	Sucrose density gradient	n/a	8 mg (12%)	[95]
*Ee*	5–20 g	Sephadex LH-20 columns	Chloroform-methanol extraction	n/a	[96]
*Ee*	3–500 g	Sepharose 2B (affinity chromatography on a CT 5263 column)	Triton X-100	3–4 mg	[93]
*Tc*	n/a	Sepharose 6B activated with cyanogen bromide (CNBr)-affinity chromatography	Triton X-100	13 mg	[97]
*Tc*	850 g	Sepharose 2B activated with CNBr -affinity chromatography	Triton X-100	30 mg (32%)	[98]
*Ee*	800 g	Affinity chromatography containing covalently bound α-cobra neurotoxin followed by cyanogen bromide affinity chromatography and, finally, ion-exchange chromatography	Tween 80	8.5%	[99]
*Ee*	3–500 g	Affinity chromatography and gel filtration using Sephadex G-75 to buffer exchange Triton X 100 to sodium cholate	Triton X-100	7–9 mg	[100]
*Tc*	150 g	Affinity chromatography α-cobratoxin-Sepharose	Triton X-100	n/a	[101]
*Tm*	n/a	Affinity chromatography neurotoxin-Sepharose 4B	Tween 20	0.4 mg/mL	[102]
*Ee*	1000 g	Affinity chromatography (Sepharose 2B) and sucrose gradients, gel filtration using Sephadex G-75 for buffer exchange	Triton X-100	0.45 mg/mL (21%)	[103]
*Tc*	600 g	Affinity chromatography	Triton X-100	115 µL/mL	[104]
*Tm*	500–1000 g	Sucrose gradient and affinity chromatography	Triton X-100	50 mg	[92]
*Tm*	800–900 g	Affinity chromatography, Sepharose 2B activated with CNBr	Triton X-100	78 mg	[105]
Sprague Dawley rats (skeletal muscle)	300–500 g (2–10 rats)	Affinity chromatography, α-cobratoxin biospecific adsorption, ion-exchange chromatography, and gel filtration steps	Triton X-100	4.6–6.0 pM	[106]
*Tc*	400 g	Affinity chromatography (α-cobratoxin) and Concanavalin A conjugated to beads to bind glycans	Sodium cholate	20%	[107]
*Tc*	60 mg of protein	Sucrose-density-gradient centrifugation and alkali treatment (pH 11.0)	Sodium cholate	^#^ 100–150 nmol	[84]
*Tc*	n/a	Sucrose density gradient and α-cobra toxin affinity chromatography	Octylglucopyranoside and Triton X-100	85–90 mg	[108]
Optic lobes from whiteleg hornchicks	n/a	Affinity chromatography using α- Bgtx-Sepharose followed by a lentil lectin gel	Triton X-100 or Lubrol PX	15–20%	[109]
*Discopyge tschudii*	200 g	Sucrose gradient and affinity chromatography on Affi-Gel 401 using bromoacetylcholine as the ligand.	Sodium cholate	2.30 mg (0.7%)	[91]
*Tm*	100–200 g	Affinity chromatography, carbachol analog ligand	Triton X-100	0.2–4.5 nmol/mg (using α-Bgtx binding)	[110]
Chick brains	300 g	Affinity chromatography: mAb 35 was coupled to Sepharose C14B	Triton X-100	0.165 mg	[111]
PC12 rat cell line	100 g	Affinity chromatography: mAb 270-Sepharose	Triton X-100	0.216–0.245 mg/100 g rat brain	[111]
*Tm*	n/a	Affinity chromatography using α-cobratoxin covalently Sepharose 2B followed by CNBr	n/a	n/a	[112]
*Tc*	100 g	Cibacron Blue Sepharose	β-octylglucopyranoside	^$^ 7 nmol	[50]
Fetal calf thymus	n/a	Affinity chromatography using α-cobratoxin-Sepharose after alkaline extraction	Triton X-100	^@^ 11.34 μg/1059 g of the thymus	[113]
*Tm*	100 g	Alkali treatment (pH 11.0), affinity chromatography (G = gallamine derivative N-(2-aminoethyl)-3,4,5-tris(2-triethylammonio-ethoxy)benzamide, C = N-(2-acetylamino- ethyl)-N-(2-aminoethyl)-N, N-dimethylammonium iodide hydroiodide, and D = N-(4-aminobenzyl)-N-dimethyl-N-(10-trimethyl-ammoniumdecyl) ammonium dichloride hydrochloride	Triton X-100	Resin C = 3.0 mg (2%), Resin D = 3.5 mg (17%), Resin G = 6.5 mg (51%)	[85]
*Tf*	120 g	Chromatofocusing, affinity chromatography using α-cobratoxin, and DEAE-Sepharose 6B	Triton X-100	Chromatofocusing = 8.3 mg (12%), affinity chromatography using cobratoxin = 3.8 mg (5.6%), and DEAE-Sepharose 6B 2.5 mg (3%)	[114]
Wistar rat brains	64 g	Affinity chromatography: DE-52 column and ACh-Affi-Gel	Lubrol PX	DE-52 = 43% and ACh-Affi-Gel 15%	[115]
Sprague Dawley rat brains	n/a	Affinity chromatography: Affi-Gel 401	Triton X-100	<0.01 mg/mL	[116]
*Tc*	120 g	Affinity chromatography: Affi-Gel 10	Triton X-100	30 mg	[117]
*Tm*	50–1000 g	Affinity chromatography: The main column employed include reversible binding of *Naja nigricollis* by lysine-15 affinity chromatography. This column was compared with the following two affinity chromatography methods: First method: α-Bgtx from *Bungarus multicinctus* polymodified in various amines groups and covalently bound to a resin. Second method: α-cobratoxin from *Naja naja kaouthia* bound to a commercial agarose resin.	CHAPS (1%)	80 pmol nAChR/mL of resin First method: 85 pmol nAChR/mL of resin 500 pmol nAChR/mL of resin	[87]
*Tc*	100 g	Affinity-purified on a bromoacetylcholine bromide-derivatized Affi-Gel 102 column	Cholate	n/a	[118]
*Tc*	5 mL	Sucrose gradient followed by affinity chromatography (using bromoacetylcholine bromide-derivatized Affi-Gel 10 column), and a PD-10 desalting column	Sodium cholate or Foscholine-12 or CHAPS or Anapoe-C12E9 or BigCHAP or Cymal-6 or DDM or LDAO or OG	n/a	[52]
*Tc*	n/a	Affinity chromatography using bromoacetylcholine bromide-derivatized Affi-Gel 10 column	Sodium cholate	n/a	[86]
*Tc*	100 g	Affinity chromatography using bromoacetylcholine bromide-derivatized Affi-Gel 102 column	Cholate	n/a	[119]
*Tc*	5 mL	Sucrose gradient followed by affinity chromatography (using bromoacetylcholine bromide-derivatized Affi-Gel 10 column), and a PD-10 desalting column	*n*-tetradecylphosphocholine, *n*-hexadecylphosphocholine f or 1-palmitoyl-2-hydroxy-sn-glycero-3-phosphocholine, 3-[(3-cholamidopropy) dimethylammonio]-2-hydroxy-1-propanesulfonate or (N,N′-bis-[3-D-gluconamidopropyl] cholamide), cymal-6, DDM, LDAO, and OG	1–5 mg	[79]
*Tm*	500–1000 g	Sucrose gradient	Triton X-100	n/a	[62]
*Tc*	60 g	Affinity purification using bromoacetylcholine bromide-derivatized Affi-Gel 10 column and a PD-10 desalting column	1-palmitoyl-2-hydroxy-sn-glycero-3-phospho-	n/a	[54,55]
*Tc*	50 g	Alkali treatment extraction at pH 11.0 followed by affinity purification (using 2-[aminobutanoyl)oxy]-N,N,N-trimethylethanaminium), and SEC	Triton X-100 and DDM	n/a	[65]
*Tc*	20–40 g	Affinity purification (using bromoacetylcholine bromide-derivatized Affi-Gel 15 column), followed by Capto lentil lectin chromatography, and gel filtration	LysoFos Choline-16 and Cholesteryl hemisuccinate	2–4 mg	[77,78]
*Tc*	50 g	Alkali treatment extraction at pH 11.0 followed by affinity purification (using 2-[aminobutanoyl)oxy]-N,N,N-trimethylethanaminium), and SEC for characterization	Triton X-100 and DDM	n/a	[120]

Abbreviations: ^1^
*Tm* = *Torpedo marmorata*, *Ee* = *Electrophorus electricus, Tc* = *Torpedo californica*, *Tf* = *Torpedo fuscomaculata*, DEAE = diethylaminoethyl cellulose, Ref. = reference, n/a = not available. Notes: ^#^ determined by the concentration of ^125^I-labeled-α-Bgtx-binding sites remaining in the supernatant, ^$^
^125^I-α-Bgtx-binding sites/mg protein, ^@^ measured by α-Bgtx binding sites per mg of protein.

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
