# Peer review of "Pursuing High-Resolution Structures of Nicotinic Acetylcholine Receptors: Lessons Learned from Five Decades"

_molecules, 2021, doi:10.3390/molecules26195753_

Round 1

Reviewer 1 Report

Please, look at the attached file

Author Response

We thank the reviewer for the comments and criticisms of this manuscript and for the opportunity to amend the issues identified. In the same way, we appreciate the attention to detail you had with this Review, without a doubt this version is superior thanks to the details and recommendations identified by you. The vast majority of the recommendations were accepted. The only one we did not include was to add an Abbreviations section since we searched the Journal webpage and it seems that it does not provide this section as part of its natural format.

Reviewer 2 Report

The paper deals with a very specific and high specialized problem. It covers a wide range of problems connected with researchs on high-resolution of acetylcholine receptors. The history of acetylcholine begins at 1867 (it was synthetized by Adolf von Bayer), through protein structure determination of nicotinic acetylcholine receptor (nACHR’s crystals) by X-ray diffraction, but does not end today, even though advanced instrumental technology has been used for its study, for example cryo-electron microscopy for nACHR’s structures. In general, this is a very interesting manuscript and it is worth recommending for publication in Molecules after minor revision.

Drawbacks of the manuscript is that it contains 20 pages of the text. Also two tables contain only the text. Moreover, there are only three figures. In my opinion the manuscript should contain more figures and tablets. Otherwise, this comprehensive review might not be interesting for the Readers.

Abbreviations: nAChRs and cry-EM should be explained in Abstract.

Author Response

We thank the reviewer for the comments and criticisms of this manuscript and for the opportunity to amend the identified weaknesses. We have accepted all your recommendations. In this revised version you will notice that more and better figures were included that will guide readers and capture their attention. We have also defined abbreviations in the Abstract following the reviewer's recommendations. We appreciate your review. 

Reviewer 3 Report

Overall this is a good review and it does a great job at summarizing key historical steps that have led us to our current understanding of nAChR structure and function. Additionally, portions of this paper are organized to very effectively show the progress of increasing structural insight regarding nAChRs (Fig 1, Table 1). Additionally, there is some high value here as there are several key issues to structure determination that I do not believe have ever been covered in a comparable review (assessment of purity, exhaustive lists of purifying methods, etc.). Despite that are some major and minor issues that make this reviewer hesitant about its readiness for publication.

Major Comments:

  1. There are several areas of this work where appropriate terminology is not utilized. For instance, sections that clearly touch upon facts regarding potency, efficacy, or affinity have 'other' words that are used instead of something that designates appropriate pharmacological principles.
  2. The section on "nAChRs as Therapeutic Targets for Neurological Diseases" is too brief. This section needs to be expanded greatly. Moreover, there is not a single reference provided in this section.
  3. As this review focuses on the evolution of nAChR structural investigation, it is disappointing to not see a figure that 'shows' the structures that have been revealed over time. I recognize this may be due to a licensing issue; but it would greatly benefit the focus of this paper. That said, the images provided Figure 2 and 3 lack an appropriate figure legend (Figure 2, no actual description of what is displayed in panels A and B, respectively) or lack key details of binding interactions (Figure 3, in the legend there is no descriptor for the identification of the alpha helices and the key amino acids discussed in the text is not highlighted). 
  4. The appropriate reference for the alpha7 cryo-EM is missing (lines 596 - 601).

Minor Issues:

  1. This review would benefit from having numbered sections. For instance, in one point, the authors use phrases like "as mentioned above" and it is very difficulty to find the previous statement that listed in detail (in this case it was the co-purifying contaminants of 'other' tissue sources). Thus the authors could benefit of adding statements like "refer to section 2.2" to provide a direction to readers.
  2. Similarly, there are numerous grammatical issues that are pervasive throughout the review.

Author Response

We thank the reviewer for the comments and criticisms of this manuscript and for the opportunity to amend the identified weaknesses. In this revised version we have considerably expanded the section "nAChRs as Therapeutic Targets for Neurological Diseases" supporting it with sufficient literature. Similarly, we have included the figure suggested by the reviewer to show the structures that have been revealed over time. In fact, this revised version includes more figures with their respective captions. Also, we have listed the sections of the manuscript, it certainly reads much better now. Finally, we have been very rigorous to eliminate grammatical errors. We appreciate the thoroughness with which you evaluated this Review.